# The Structure–Property Relationship of Pyrrolidinium and Piperidinium-Based Bromide Organic Materials

**DOI:** 10.3390/ma15238483

**Published:** 2022-11-28

**Authors:** Claudio Ferdeghini, Andrea Mezzetta, Felicia D’Andrea, Christian Silvio Pomelli, Lorenzo Guazzelli, Luca Guglielmero

**Affiliations:** 1Department of Pharmacy, University of Pisa, Via Bonanno 33, 56126 Pisa, Italy; 2Scuola Normale Superiore, Piazza dei Cavalieri 7, 56126 Pisa, Italy

**Keywords:** ionic liquids (ILs), dicationic ionic liquids (DILs), thermogravimetric analysis (TGA)

## Abstract

Two couples of dicationic ionic liquids, featuring pyrrolidinium and piperidinium cations and different linker chains, were prepared and characterized. 1,1′-(propane-1,3-diyl)bis(1-methylpyrrolidinium) bromide, 1,1′-(octane-1,8-diyl)bis(1-methylpyrrolidinium) bromide, 1,1′-(propane-1,3-diyl)bis(1-methylpiperidinium) bromide, and 1,1′-(octane-1,8-diyl)bis(1-methylpiperidinium) bromide were synthesized in quantitative yields and high purity and thermally characterized through TGA and DSC analysis. In this study, we propose a preliminary comparative evaluation of the effect of the linker chain length and of the size of the aliphatic ammonium ring on the thermal and solubility properties of bromide dicationic ionic liquids.

## 1. Introduction

Ionic liquids (ILs) are a class of low melting organic salts, which have attracted a great amount of research interest in the past two decades [1]. Due to their peculiar physicochemical properties, including negligible vapor pressure, good thermal stability, high ionic and thermal conductivity, low flammability, wide electrochemical window, and unique solvent properties [2,3,4,5,6,7,8], ILs have found broad use as catalysts and as unconventional solvents for synthesis [9,10,11,12], as well as for a wide variety of different applications, spanning from electrochemical applications [13,14,15,16,17], biomass processing [18,19,20], preparation of carbon materials [21], and CO_2_ capture [22], to drug delivery and utilization as potential drug components [23,24,25].

The great variety of different anions and cations, together with the possibility to functionalize their structure for particular applications, makes ILs versatile and flexible systems. Specific structures can be designed to suit the requirements for a specific application. The introduction of dicationic structures [26], with two positively charged groups connected by a linker chain, further expands the tunability potential of the physicochemical properties, by acting on the length of the linker chain as well as on possible asymmetries on the structure. Task specific dicationic ionic liquids (DILs) have been designed and proposed for several applications, where superior thermal stability, higher electrochemical windows, higher densities or larger surface tensions are required [26,27,28,29,30]. DILs have been successfully used in CO_2_ capture [31], tribology [32], catalysis in organic reactions [33], zeolites preparation [34], agriculture [35], and electrochemical application [35].

The operational window for DILs is typically between the melting and degradation temperatures. Indeed, due to their very low vapor pressure they degrade before boiling. Unfortunately, the data reported in the literature are scattered, and the measuring conditions are often different. This latter issue renders a comparison of the operating ranges of DILs difficult and sometimes impossible.

In this context, we report the preparation of two couples of dicationic systems, featuring pyrrolidinium and piperidinium cations and different linker chains. The prepared compounds were characterized by ^1^H-NMR, ^13^C-NMR, ATR-FTIR, and mass spectroscopy. The relationship between the structural features of the prepared salts and their physicochemical properties was investigated. The effect of the linker chain length and of the type of cationic heads on the solubility of the tested salts in a panel of commonly used solvents was determined. Finally, the thermal properties of all the prepared dicationic systems were studied through differential scanning calorimetry and thermogravimetric analysis, providing a preliminary comparative investigation on the effect of both the length of the linker chain and the size of the aliphatic ammonium ring on the thermal behavior and stability of this set of compounds.

## 2. Results

The desired compounds 1,1′-(propane-1,3-diyl)bis(1-methylpyrrolidinium) bromide, (**1**, [C_3_Mpyrr_2_]2Br), 1,1′-(propane-1,3-diyl)bis(1-methylpiperidinium) bromide (**2**, [C_3_Mpip_2_]2Br), 1,1′-(octane-1,3-diyl)bis(1-methylpyrrolidinium) bromide (**3**, [C_8_Mpyrr_2_]2Br), and 1,1′-(octane-1,3-diyl)bis(1-methylpiperidinium) bromide (**4**, [C_8_Mpip_2_]2Br) were prepared according to the synthetic procedure reported below (Figure 1).

The desired compounds were obtained in a one-step reaction, adopting reaction conditions similar to the ones reported in a previous work of our research group [36]. *N*-methylpyrrolidine and *N*-methylpiperidine were used as nucleophiles for the Menshutkin reaction alternatively with 1,3-dibromopropane and 1,8-dibromooctane, in order to obtain two sets of DILs with different linker chains. The four reactions were conducted in CH_3_CN at 80 °C for 48 h, obtaining the DILs as white solids in good to quantitative yields. The excess of unreacted precursors was removed by washing the solids with diethyl ether. The remaining traces of the washing solvent were finally removed under vacuum at 80 °C for 2 h. No further purification processes were needed.

^1^H-NMR, ^13^C-NMR, DEPT-135 NMR, and ESI-MS experiments confirmed the structures and the purities of compounds **1**–**4** (Appendix A). DEPT-135 NMR spectra (Appendix A) were performed to detect the signal of the methyl group, otherwise covered by the solvent residual signal of CD_3_OD in the ^13^C-NMR (Appendix A) of compounds **1**–**4**. The solubility of the prepared DILs was evaluated in a panel of solvents (Table 1). From the obtained results, there emerged a strong dependence of the solubility characteristics from the compound structure. Pyrrolidinium-based salts (**1** and **3**) displayed a higher solubility in methanol, ethanol, 2-propanol, and acetonitrile, compared to the related piperidinium salts, with the strongest difference between compounds **1** and **2**. [C_3_Mpyrr_2_]2Br (**1**) exhibited an extremely high solubility in methanol (2.526 M) and ethanol (1.470 M), while [C_3_Mpip_2_]2Br (**2**) displayed solubility values less than one fourth in methanol and even below the detectable solubility of the test for 2-propanol and acetonitrile. From the comparison between the solubility values obtained for [C_3_Mpyrr_2_]2Br (**1**) and [C_8_Mpyrr_2_]2Br (**3**), the elongation of the linker chain provided a general reduction in the solubility in all the tested organic solvents, reducing by about half the values observed for compound **1**. On the other hand, passing from a C_3_ to a C_8_ linker chain for the piperidinium based systems led to an increase in the solubility, which was found to still be lower for **4** than its pyrrolidinium counterpart **3** but much higher than compound **2** for ethanol, 2-propanol, and acetonitrile. The other tested organic solvents, acetone, ethyl acetate, and dichloromethane were found to be unsuitable for the dissolution of all the tested compounds. On the other end of the scale, all the presented salts displayed an extremely high solubility in water, where only few drops of this solvent were sufficient to fully dissolve the whole sample. In these cases, it was not possible to achieve a precise determination of the solubility since no biphasic systems were obtainable, and the smallest amount of water was already enough to turn the samples into a glassy gluey body. The excellent water solubility observed for the considered salts as well as their insolubility in apolar organic solvents is consistent with the data reported by Anderson et al. for compound **1** [26,37].

The extremely sharp differences in solubility due to the small structural differences existing between the tested compounds, seen in these solubility tests, represents an interesting result for the study of the structure–property relationship of this class of dicationic systems.

The thermal behavior of [C_3_Mpyrr_2_]2Br (**1**), [C_3_Mpip_2_]2Br (**2**), [C_8_Mpyrr_2_]2Br (**3**), and [C_8_Mpip_2_]2Br (**4**) was investigated through thermal gravimetric analysis (TGA) and differential scanning calorimetry (DSC). The thermal stability was assessed by TGA, heating from 40 to 700 °C with a heating rate of 10 °C min^−1^ under a nitrogen atmosphere using a platinum crucible. The T_peak_, T_onset_, and T_start 5%_ temperatures obtained from the thermograms (Appendix A) are reported in Table 2.

From the data reported in Table 2, no clear effects on the thermal stability of the tested compounds were appreciable for both the linker chain length and the type of cationic group. For the T_start 5%_ values, a not intense but still visible difference in thermal stability was appreciated between the short linker chains DILs and the long linker chain ones, with the latter starting their decomposition at slightly higher temperatures. On the other hand, the T_onset_ and T_peak_ values did not confirm this trend and appeared substantially independent from the structural parameters studied in this test. The obtained results are coherent with the ones reported in the literature for similar morpholine-based DILs bromide [30], where minimum differences were observed for the degradation temperatures of DILs with same cationic heads (Figure 1). Higher T_peak_s were recorded for the tested pyrrolidinium and piperidinium compounds **1**–**4** with respect to the similar dicationic morpholinium systems reported by our research group in a previous work [30], showing that these cationic heads can provide more thermally stable DILs, with values almost matching those reported in the literature for imidazolium DILs [36]. More interesting differences were displayed on the thermal degradation profiles. From the dweight/dT plots of [C_3_Mpip_2_]2Br (**2**), the degradation process, even if basically occurring in a single step, was still distinct into two almost coincident separated degradations (Figure 1, Appendix A), Differently, this feature was not appreciable in the dweight/dT plots of [C_8_Mpip_2_]2Br (**4**), where the two main processes were so close in temperature as to be undistinguishable (Figure 1, Appendix A). The observed behavior was consistent with the degradation mechanism described for the related morpholinium DILs bromide [30], even though in the present case, the double step degradation appeared much less pronounced.

Differential scanning calorimetry (DSC) analysis of compounds **1**–**4** was also performed (Figure 2), and the obtained data are reported in Table 2. Compound **1** displayed a solid–solid transition during the cooling run subsequent to the preliminary drying step (endothermic peak at 34.2 °C, 1.18 kJ mol^−1^) and a corresponding solid–solid transition during the next heating run, characterized by both similar temperature and enthalpy (exothermic peak at 30.7 °C, 1.07 kJ mol^−1^) (Figure 2a). A second thermal phenomenon was observed during the same heating run at 209.2 °C, identified as a melting transition. No crystallization was observed during the subsequent cooling run, and no solid–solid or melting transitions were detected in the next steps of the DSC analysis. The obtained results were in contrast to the ones reported in the literature by Anderson et al. relative to the same compound, where a low melting point was reported [26]. The melting point measure during this work was instead found to be about 30 °C higher than the one measured for the related long-chained compound **3**, which was consistent with the general melting point trend reported in the literature for different dicationic systems in relation to the length of their linker chain [26,36,38]. The different results could be due to the different water contents of the samples. For compound **2**, no thermal transition was detectable up to 260 °C, where only a small exothermic signal associated with the beginning of the thermal degradation (T_start 5%_ 272.3 °C) was observed (Appendix A). Compound **3**, sharing with **1** the presence of pyrrolidinium cationic groups, displayed a melting transition at 174.3 °C while, similar to the related compound **1**, no crystallization was observed during the subsequent cooling step (Figure 2b). As already observed for the C_3_ linker chain piperidinium salt, [C_8_Mpip_2_]2Br (**4**) was found to not undergo any melting process up to 260 °C (Appendix A). On the other hand, solid–solid transitions were detected both during the first and the second heating–cooling cycle with an exothermic phenomenon during the 1^st^ heating run at 141.4 °C (3.00 KJ mol^−1^) and a related endothermic one during the 1^st^ cooling run at 135.8 °C (3.08 KJ mol^−1^) (Figure 2c). Only minimum differences in terms of temperature and associated enthalpy were appreciable between the two cycles at 10 °C min^−1^, and the same peak temperatures were also found for a supplemental cycle performed at 5 °C min^−1^ (Appendix A). The identification of these peaks as solid–solid transitions was confirmed by the use of the Kofler technique, which clearly showed no melting transitions up to above 200 °C (Appendix A).

For ILs, the operative range defines the interval of temperatures where the studied system is liquid and usable for applications. It is usually determined by considering the degradation temperature obtained from the TGA as the upper utilization limit and the melting point measured by DSC as the lowest limit. Even though not classifiable as an ionic liquid with full rights due to its high melting point, an operative range of 106.0 °C was indicated for compound **3** (considering the upper limit as the T_start 5%_ value). It is important to notice that for a meaningful comparison of operative ranges of different systems, thermal analyses need to have the same experimental settings. In conclusion, both the length of the linker chain and the type of cation heads represented an influential element in the determination of the thermal behavior of the considered salts, with the pyrrolidinium C_8_ linker compounds melting at a sensibly higher temperature with respect to the solid–solid transition observed for its related piperidinium compound, and the C_3_ systems, on the other hand, not manifesting any detectable thermal transition.

## 3. Materials and Methods

### 3.1. General Information

NMR spectra were recorded with a Bruker Avance II (Bruker Italia S.r.l., Milano, Italy) operating at 400 (^1^H) and 100 MHz (^13^C). The first-order proton chemical shifts were referenced to the residual CD_3_OD (δ_H_ 3.31, δ_C_ 49.03). The chemical shifts were given in δ. The following abbreviations were used; s = singlet, m = multiplet.

ATR-FTIR spectra were recorded with an IR Cary 600 FTIR spectrometer (Agilent Technologies, Santa Clara, CA, USA) using a macroATR accessory with a diamond crystal. The spectra were measured in a range from 4000 to 500 cm^−1^, with 32 scans both for background and samples.

High resolution mass spectra were acquired on an Orbitrap Q Exactive Plus with H-ESI source (Thermo Fischer Scientific Inc., Bremen, Germany) in ESI positive ionization mode operating with the following parameters: a spray voltage of 3400 V, capillary temperature of 290 °C, S-lens RF level 50, sheat gas (N_2_) 24 arbitrary unit, and auxiliary gas (N_2_) 5 arbitrary unit. The data were elaborated with Xcalibur software.

All reagents and solvents were obtained from Merck Life Science S.r.l. (Darmstadt, Germany) or Thermo Fisher (Germany) and used without further purification.

The thermal stability of the synthesized ILs was investigated by thermal gravimetric analysis (TGA), using a TA Instruments Q500 TGA (TA Instruments, New Castle, DE, USA). The temperature calibration was performed using a nickel standard, and for weight calibration, we used weight standards (1 g, 500 mg, and 100 mg). All standards were supplied by TA Instruments Inc. The ILs (5–15 mg) were heated in a platinum crucible as a sample holder from 40 °C to 700 °C at 10 °C/min under nitrogen (90 mL/min). The TGA experiments were carried out in triplicate.

The thermal behavior of the ionic liquids was analyzed by a differential scanning calorimeter TA Instruments DSC, Q250 (TA Instruments, New Castle, DE, USA). Dry high-purity N_2_ gas with a flow rate of 30 mL/min was purged through the sample. The sample (2–5 mg) was loaded in hermetic aluminum crucibles and dried at 130 °C for 30 min. Then, the phase behavior was explored under nitrogen atmosphere in the temperature range of −90 to 260 °C, depending on the analyzed compound. The enthalpy was also calibrated using indium (melting enthalpy ∆_m_*H* = 28.71 J/g). The DSC experiments were carried out in duplicate.

The solubility of compounds **1**–**4** in a panel of selected solvents was determined by a gravimetric approach, at room temperature. To the selected compound a quantity of solvent was added, enough to retain some solid residue, in order to obtain a saturated solution. The mixture was stirred for 10 min at reflux temperature and then for 2 h in a bath at 25 °C. If the solid was found to be completely solubilized, more salt was added, and the dissolution procedure was repeated. A carefully measured volume of the decanted solution was taken and dried under vacuum for 2 h at 80 °C. The ratio between the obtained mass and the taken volume was used to calculate the concentration of the selected compound in the solvent at the saturation point.

### 3.2. General Procedure for the Synthesis of Compounds ***1***–***4***

Compounds **1**–**4** were obtained following a general procedure previously reported by our group [36]. In a flask, to a solution of 2.1 equiv (15 mmol) of *N*-methyl heterocycle in 5 mL of CH_3_CN, a solution of 1 equiv of 1,n-dibromoalkane in 15 mL of CH_3_CN was added dropwise. The reaction mixture was then heated and stirred at 80 °C for 48 h. The solvent was removed under reduced pressure, and the obtained solid was washed with Et_2_O (3 × 50 mL); then, it was dried under reduced pressure affording the desired product in quantitative yield as a white solid.

#### 3.2.1. Synthesis of Ionic Liquid 1 [C_3_Mpyrr_2_]2Br

The preparation of **1** (97% yield, hygroscopic white solid) was performed according to the general procedure. ^1^H-NMR (CD_3_OD) δ: 3.66 (m, 8H, 4 × N*CH_2_*CH_2_ pyrrolidinium), 3.52 (m, 4H, 2 × N*CH_2_*CH_2_ linker chain), 3.20 (s, 6H, 2 × N*CH_3_*), 2.43 (m, 2H, NCH_2_*CH_2_*CH_2_N), and 2.28 (m, 8H, NCH_2_*CH_2_* pyrrolidinium). The obtained spectrum was in agreement with the literature data [26]. Minor variations in the chemical shift were ascribable to the different deuterated solvent used. ^13^C-NMR (CD_3_OD) δ: 66.0 (N*CH_2_*CH_2_ pyrrolidinium), 61.5 (N*CH_2_*CH_2_ linker chain), 49.5 (N*CH_3_*), 22.7 (NCH_2_*CH_2_* pyrrolidinium), and 20.7 (NCH_2_*CH_2_*CH_2_N linker chain). ATR-FTIR (cm^−1^): 3001, 2959, 2888, 1456, 1420, 1373, 1306, 1054, 999, 930, 911, 827, and 763. ESI-MS positive mode calculated for [C_13_H_28_N_2_]^2+^: *m*/*z* = 106.1121 (z = 2); found: *m*/*z* = 106.1126 (z = 2).

#### 3.2.2. Synthesis of Ionic Liquid 2 [C_3_Mpip_2_]2Br

The preparation of **2** (86% yield, hygroscopic white solid) was performed according to the general procedure. ^1^H-NMR (CD_3_OD) δ: 3.54 (m, 12H, 6 × N*CH_2_*CH_2_ piperidinium and linker chain), 3.21 (s, 6H, 2 × N*CH_3_*), 2.36 (m, 2H, NCH_2_*CH_2_*CH_2_N linker chain), 1.97 (m, 8H, 4 × NCH_2_*CH_2_* piperidinium), and 1.73 (m, 4H, 2 × NCH_2_CH_2_*CH_2_* piperidinium). ^13^C-NMR (CD_3_OD) δ: 62.8 (N*CH_2_*CH_2_ piperidinium), 60.9 (N*CH_2_*CH_2_ linker chain), 50.0 (N*CH_3_*), 22.0 (NCH_2_CH_2_*CH_2_* piperidinium), 21.0 (NCH_2_*CH_2_* piperidinium), and 17.1 (NCH_2_*CH_2_*CH_2_N linker chain). ATR-FTIR (cm^−1^): 2997, 2966, 2946, 2903, 2858, 1490, 1468, 1445, 1403, 1349, 1329, 1278, 1234, 1196, 1112, 1058, 1037, 996, 970, 944, 912, 889, 867, 825, 795, and 755. ESI-MS positive mode calculated for [C_15_H_32_N_2_]^2+^: *m*/*z* = 120.1277 (z = 2); found: *m*/*z* = 120.1278 (z = 2).

#### 3.2.3. Synthesis of Ionic Liquid 3 [C_8_Mpyrr_2_]2Br

The preparation of **3** (99% yield, hygroscopic white solid) was performed according to the general procedure. ^1^H-NMR (CD_3_OD) δ: 3.58 (m, 8H, 4 × N*CH_2_*CH_2_ pyrrolidinium), 3.43 (m, 4H, 2 × N*CH_2_*CH_2_ linker chain), 3.11 (s, 6H, 2 × N*CH_3_*), 2.24 (m, 8H, 4 × NCH_2_*CH_2_* pyrrolidinium), 1.84 (m, 4H, 2 × NCH_2_*CH_2_* linker chain), and 1.46 (m, 8H, 2 × NCH_2_CH_2_*CH_2_* and 2 × NCH_2_CH_2_CH_2_*CH_2_* linker chain). ^13^C-NMR (CD_3_OD) δ: 65.5 (N*CH_2_*CH_2_ linker chain), 65.4 (N*CH_2_*CH_2_ pyrrolidinium), 48.9 (N*CH_3_*), 29.7 (NCH_2_CH_2_CH_2_*CH_2_* linker chain), 27.2 (NCH_2_CH_2_*CH_2_* linker chain), 24.7 (NCH_2_*CH_2_* linker chain), and 22.6 (NCH_2_*CH_2_* pyrrolidinium). ATR-FTIR (cm^−1^): 3008, 2937, 2856, 1473, 1455, 1413, 1310, 1049, 1000, 956, 930, 837, and 729. ESI-MS positive mode calculated for [C_18_H_38_N_2_]^2+^: *m*/*z* = 141.1512 (z = 2); found: *m*/*z* = 141.1514 (z = 2).

#### 3.2.4. Synthesis of Ionic Liquid 4 [C_8_Mpip_2_]2Br

The preparation of **4** (97% yield, hygroscopic white solid) was performed according to the general procedure. ^1^H-NMR (CD_3_OD) δ: 3.42 (m, 12H, 6 × N*CH_2_*CH_2_ piperidinium and linker chain), 3.10 (s, 6H, 2 × N*CH_3_*), 1.92 (m, 8H, 4 × NCH_2_*CH_2_* piperidinium), 1.60–1.85 (m, 8H, 4 × NCH_2_*CH_2_* linker chain and 4 × NCH_2_CH_2_*CH_2_* piperidinium), and 1.46 (m, 8H 4 × NCH_2_CH_2_*CH_2_* and 4 × NCH_2_CH_2_CH_2_*CH_2_* linker chain). ^13^C-NMR (CD_3_OD) δ: 65.1 (N*CH_2_*CH_2_ piperidinium), 62.2 (N*CH_2_*CH_2_ linker chain), 48.2 (N*CH_3_*), 29.8 (NCH_2_CH_2_CH_2_*CH_2_* linker chain), 27.2 (NCH_2_CH_2_*CH_2_* linker chain), 22.7 (NCH_2_*CH_2_* linker chain), 22.1 (NCH_2_CH_2_*CH_2_* piperidinium), and 21.0 (NCH_2_*CH_2_* piperidinium). ATR-FTIR (cm^−1^): 2927, 2876, 2853, 1477, 1460, 1447, 1401, 1329, 1273, 1235, 1200, 1080, 1033, 983, 946, 911, 884, 865, 729, and 606. ESI-MS positive mode calculated for [C_20_H_42_N_2_]^2+^: *m*/*z* = 155.1669 (z = 2); found: *m*/*z* = 155.1670 (z = 2).

## 4. Conclusions

In conclusion, four bromide dicationic organic salts were synthesized in high purity and quantitative yields. All the compounds were characterized through ESI-MS, ^1^H-NMR, ^13^C-NMR, DEPT-135 NMR, and ATR-FTIR, their solubility was measured in a panel of commonly used solvents, and their operative range was investigated by TGA and DSC techniques. The thermogravimetric analysis showed a good thermal stability both for pyrrolidinium and piperidinium salts, comparable to the one exhibited by imidazolium DILs, and a substantial independence of the thermal stability from the linker chain. The thermal behavior was, on the other hand found, to be sharply influenced by the structure of the studied compounds. The pyrrolidinium salts displayed high melting temperatures, while the piperidinium compounds were found to degrade before the melting point. The effect of the linker chain was also appreciable with a lower melting point reported for the dicationic system with a longer linker chain, coherent with the behavior manifested by other classes of dicationic ionic liquids. As a result of this study, the investigated compounds should more correctly be referred to as dicationic organic salts, due to their high melting temperatures. Nevertheless, the use of compounds **1**–**4** as precursors for the preparation of salts featuring the same dicationic structure but different anions can predictably lead to the formation of proper dicationic ionic liquid systems. Moreover, the solubility study evidenced a remarkable difference between the related pyrrolidinium and piperidinium dicationic salts, with the latter being less soluble in the whole panel of tested solvents. This characteristic appears to be amplified by the short linker chains, with compounds **1** and **2** respectively displaying the highest and the lowest observed solubility for each solvent.

The results obtained in this study provided some interesting insights on the presence of a strong relationship between the structural and physicochemical properties in pyrrolidinium and piperidinium-based dicationic systems.

## Data Availability

Not applicable.

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
