# Peer review of "The Structure–Property Relationship of Pyrrolidinium and Piperidinium-Based Bromide Organic Materials"

_materials, 2022, doi:10.3390/ma15238483_

Round 1

Reviewer 1 Report

The article submitted to me for review concerns the synthesis and physicochemical characterisation of bromide dicationic ionic liquids. The abstract is concise and to the point.

The introduction of the paper is correct. The uses of ionic liquids are described very generally. Unfortunately, the ionic liquids synthesised by the authors will not be applicable in many of them because they are solid at T = 298 K and p = 1 atm. Moreover, they have high melting points.

The synthesis scheme is identical for the four ionic liquids obtained by the authors. The synthesis pathway is known from publications and is very simple. In my opinion, it is too little for publication. The authors obtained bromide organic salts, precursors of ionic liquids. By definition, an ionic liquid should melt below 373.15 K (100oC). Here the melting points are much higher. They are therefore organic salts. Their quality and purity are determined by NMR and ESI-MS spectroscopy. This could be supplemented by FTIR spectra.

In my opinion, the solubility test does not contribute much. We only know whether or not 0.1 g of salt will dissolve in a given solvent. At what temperature? Salt solubility is a function of temperature. There are missing solid-liquid phase equilibria diagrams from which the solubility limits can be determined. These can be determined, for example, by the DSC technique used by the authors.

The biggest problem is the results from differential scanning calorimetry, DSC or lack thereof. Why did the authors perform measurements in the range -100oC to 200oC ? The salts are solid in this temperature range. Of course, a solid-solid phase transition can be expected (the authors have indicated this), but the melting point is missing. Melting is above 200oC, except for the salt [C8Mpyrr2]2Br. For it, the melting point is 174oC. The Q250 calorimeter allows heating to above 400oC. Of course, to avoid decomposition of the sample, it suggests a range up to 270oC. Surely you can catch the melting just before the sample decomposes.

The manuscript in its present form is not enough for publication. It could be expanded to include solid-liquid phase diagrams over the full concentration range. Synthesis of ionic liquids with different anions based on these salts can be added. Certainly, the DSC results for a wider range of T need to be repeated, and the melting point and enthalpy of fusion of the other salts are determined.

Errors:

Names of the ionic liquids (organic salts)– octane-1,8-diyl (Page 1 line 14-17)

Page 2 line 75. I would disagree with the sentence: "No further purification processes were needed". It is necessary to remove the diethyl ether solvent under a vacuum at a higher temperature.

Table 1 – lack of conditions, measurement temperature.

Tabel 1 – abbreviations are not explained. Why does acetone not have an abbreviation?

Tabel 2 – kJ/mol but not KJ/mol

Reviewer 2 Report

I have received the Manuscript entitled: ‘Synthesis and characterization of pyrrolidinium and piperidinium-based bromide dicationic ionic liquids’ (Manuscript ID: materials-2008119)  submitted to the Materials for a review.

The Manuscript describes a very interesting research regarding synthesis and analysis of dicationic ionic liquids (ILs) functionalized with pyrrolidinium and piperidinium groups. Indeed, Among dicationic salts, compounds comprising conventional alkylammonium or imidazolium moiety are the most popular, which justifies the need to test other compounds, such us salts proposed by authors of the submitted Manuscript. The authors proved that they are able to obtain new ILs characterized by great purity (confirmed with the use of various spectral techniques). The procedures described in the Manuscript (synthesis, thermal analysis) are very well developed by the authors and do not raise any objections in this matter. Moreover, the discussion in the paper is well-founded and the graphical representation of the gathered data is of high quality. I consider the reviewed Manuscript as a good candidate for publication, however, there are some elements that require improvement or further explanation:

·       In supporting information Caption of Figure S8. Should start from 13C NMR instead of 1H-NMR

·       General procedure for analysis of solubility is missing. This part should also contain explanation of the abbreviations of solvents provided in the table 1

·       Symbols Tpeak, Tonset and Tstart 5% should be explained below the table 2 or in its caption

·       In the sentence “From dweight/dT plots of [C3Mpip2]2Br (2) it is visible that 117 the degradation process, even if basically happening in a single step, can still be distinct 118 into two almost coincident separated degradations (Figure 1, Figures S16 see supporting 119 information),” authors should refer to figure S14 as it contains thermogram of compound 2

·       Similarly, in the next sentence there should be Figure S16 instead of S18

·       The authors wrote “On the other hand [C8Mpip2]2Br (4) displayed solid-solid transitions 132 characterized by low enthalpies, with an exothermic phenomenon during the heating 133 phase at 139.4 °C (9.670 KJ mol-1) and a related endothermic one during the cooling phase 134 at 133.6 °C (9.689 KJ mol-1).” However, how is was established that it is a solid-solid transition instead of basic melting and subsequent crystallization? E.g. various scan rates are being applied for this purpose, but I was unable to find such information

·       Conclusions should be strengthen with 2-3 sentences and provide more information, particularly the DSC/TG analyses should be described with more details

Reviewer 3 Report

This paper reports the synthesis and characterzation of 4 dicationic ionic liquids (DILs), pyrrolidinium and piperidinium, with two different spacers, i.e. a 3 and a 6 alkyl chain.

In the paper the synthesis, NMR, ESI-MS, TGA and DSC characterization is reported, along with the solubility study in most common solvents.

The effect of the length of the spacer does not seem very important in solubility and thermal characteristics. No degradation mechanism is proposed.

I see some criticisms to be addressed before publication.

In particular, the synthesis, NMR, MS and thermal characterization of some of the 4 reported DILs is reported in the literature (J. AM. CHEM. SOC. 2005, 127, 593-604; Journal of Catalysis 215 (2003) 151–170; US Patent US 2006/0025598 A1) and must be cited and a comparison with the results reported.

As regards the solubility, J. AM. CHEM. SOC. 2005, 127, 593-604  reports the miscibility of 1 with hexane and water and again a comparison should be reported.

In Table 1 the solubility of the 4 DILs is reported as a yes/no report. In the caption it is written that a DIL is consideed soluble if the solubility is above 0.1 mg/mL, which means for 4 0.23 mM. Not so soluble. Moreover, in the experimental part of the paper the solubility measure process is not reported. I think that it would be better if the solubility values are reported, in order to distinguish between DILs and to understand the real cotent of the solutions.

Reviewer 4 Report

The authors synthesized a series of different types of ionic liquids with double cationic groups and different carbon number links. The molecular structures were corroborated by relevant characterization and the melting points and degradation temperatures of the compounds were confirmed using thermogravimetric and differential thermal analysis.

Overall, the article successfully synthesized the target ionic liquids and provided information on their associated window temperatures. I suggest that the article undergoes some modifications prior to publication with the following information.

(1) Could the authors please explain what is the significance of the currently designed molecular structure? Is it believed that the molecular structure can greatly improve the temperature range of ionic liquids or something else?

(2) After reading the article, I have been unable to understand whether the authors want to emphasize the innovation in the synthesis of new compounds or the determination of the temperature range of use?

(3) What new insights does the article give into the field of ionic liquids?

Round 2

Reviewer 4 Report

I accept the author's corrections to the article and recommend it for publication.